# Utility of smart watches for identifying arrhythmias in children

Aydin Zahedivash [1], Henry Chubb[1], Heather Giacone [1], Nicole K. Boramanand[1], Anne M. Dubin[1], Anthony Trela[1], Erin Lencioni[1], Kara S. Motonaga[1], William Goodyer[1], Brittany Navarre[1], Vishnu Ravi [2], Paul Schmiedmayer [2], Vasiliki Bikia [2], Oliver Aalami [2], Xuefeng B. Ling[3], Marco Perez [4] & Scott R. Ceresnak[1✉]

## Abstract

**Background** Arrhythmia symptoms are frequent complaints in children and often require a pediatric cardiology evaluation. Data regarding the clinical utility of wearable technologies are limited in children. We hypothesize that an Apple Watch can capture arrhythmias in children.

**Methods** We present an analysis of patients ≤18 years-of-age who had signs of an arrhythmia documented by an Apple Watch. We include patients evaluated at our center over a 4-year-period and highlight those receiving a formal arrhythmia diagnosis. We evaluate the role of the Apple Watch in arrhythmia diagnosis, the results of other ambulatory cardiac monitoring studies, and findings of any EP studies.

**Results** We identify 145 electronic-medical-record identifications of *Apple Watch*, and find arrhythmias confirmed in 41 patients (28%) [mean age 13.8 ± 3.2 years]. The arrythmias include: 36 SVT (88%), 3 VT (7%), 1 heart block (2.5%) and wide 1 complex tachycardia (2.5%). We show that invasive EP study confirmed diagnosis in 34 of the 36 patients (94%) with SVT (2 non-inducible). We find that the Apple Watch helped prompt a workup resulting in a new arrhythmia diagnosis for 29 patients (71%). We note traditional ambulatory cardiac monitors were worn by 35 patients (85%), which did not detect arrhythmias in 10 patients (29%). In 73 patients who used an Apple Watch for recreational or self-directed heart rate monitoring, 18 (25%) sought care due to device findings without any arrhythmias identified.

**Conclusion** We demonstrate that the Apple Watch can record arrhythmia events in children, including events not identified on traditionally used ambulatory monitors.

## Plain language summary

Wearable devices, such as smart watches, have become popular for the monitoring of health, particularly for people with heart conditions. Wearable devices have been well-studied in adults, however there is less information available on their effectiveness in monitoring children's health. We reviewed the heart electrical recordings of a group of children who submitted recordings obtained from their Apple Watches during moments when they felt as though their heart's rhythm was abnormal. The Apple Watches captured rhythm abnormalities that matched the diagnoses obtained using heart monitors used clinically. This study shows that use of Apple Watches can enable clinicians to identify abnormalities that many traditional at-home monitoring devices do not detect. Thus, wearable devices, such as the Apple Watch, could be used to help identify heart rhythm disorders in children.

[1] Stanford University, Lucile Packard Children's Hospital, Department of Pediatrics, Pediatric Cardiology, Palo Alto, CA, USA. [2] Stanford University, Stanford Byers Center for Biodesign, Palo Alto, CA, USA. [3] Stanford University, Department of Surgery, Palo Alto, CA, USA. [4] Stanford University, Cardiovascular Medicine – Electrophysiology, Department of Medicine, Palo Alto, CA, USA. ✉email: ceresnak@stanford.edu

Palpitations and concern for heart rhythm abnormalities are among the most common causes of referral for subspecialty care in pediatric cardiology[1–3]. Workup of a potential pediatric arrhythmia requires ambulatory cardiac rhythm monitoring, and current noninvasive technologies carry important limitations. Current diagnostic options for children include patch rhythm monitors, 30-day event monitors, and 24–48-h Holter monitors. These options may not be long enough to capture a patient's symptoms due to multiple factors, including a shorter wear time in children compared to adults, skin irritation limiting use, and the often sporadic nature of symptoms[4,5]. Longer-term monitoring is possible with an implantable loop recorder (ILR)[6], but involves an invasive procedure. There remains a clear need for longer-term noninvasive extended cardiac monitoring in children.

The Apple Watch has been shown in adults to detect a variety of arrhythmias[7,8], and wearable monitors are being recognized as increasingly valuable diagnostic tools within digital health[9]. Wearable technologies carry unique challenges in children compared to adults, however, as children have notably higher heart rates, higher activity levels, and current arrhythmia detection algorithms have not been designed for children or the types of arrhythmias most commonly seen in this population. The Apple watch has two principal features for heart monitoring: passive heart rate alerts from an optical sensor, and patient-initiated ECG recordings using its electrical sensors. There are limited data in the pediatric literature assessing the reliability of several Apple Watch electrocardiogram (ECG) data points (such as heart rate, intervals, and amplitudes) in children, but only limited data on arrhythmia characterization and use of Apple Watch for extended heart rhythm monitoring in children[10–13].

We investigate the utility of the Apple Watch for arrhythmia characterization in children and hypothesize that the Apple Watch could be used to help identify arrhythmias in children. We show that the Apple Watch is able to capture clinical arrhythmia events in children, and for many patients with confirmed arrhythmia (71%) the Apple Watch prompted them to present to care which resulted in a new arrhythmia diagnosis. Additionally, we show that the Apple Watch captured arrhythmia events in children where conventional ambulatory cardiac rhythm monitors did not. We show that wearable devices such as the Apple Watch can capture arrhythmia events in children and justify the need for further investigation regarding their diagnostic utility as ambulatory event monitors.

## Methods

**Study design**. A single center, retrospective analysis was performed for all patients ≤18 years of age who had signs of an arrhythmia documented by an Apple Watch between 2018 and 2022 and received a formal arrhythmia diagnosis at Lucille Packard Children's Hospital Stanford.

**Patient sample**. Patient documentation was queried for any mention of the key phrase *Apple Watch*. All data was voluntarily submitted and originated from patient- or caregiver-owned Apple Watches and no patient was asked to purchase any additional equipment. All patients with an arrhythmia captured by their Apple Watch with an arrhythmia confirmation by a pediatric electrophysiologist were included in the analysis.

**Data elements**. Data collected included: basic patient demographics, type of arrhythmia identified, the role that the Apple Watch had in arrhythmia diagnosis, the results of other ambulatory cardiac monitoring studies, interventions based on Apple Watch data, and findings of any invasive electrophysiology (EP) studies. The study protocol was approved by the Stanford University Institutional Review Board (IRB) and the need for informed consent from guardians was waived by the IRB.

Patient-submitted data included passive heart rate alerts from measurements collected by the Apple Watch optical heart rate sensor or patient-initiated ECG recordings from the device's electrical sensors (Apple Watch Series 4 and up). Electrocardiogram recordings using the Apple Watch were initiated by opening the ECG app, resting the watch-bearing arm on a steady surface, and touching a finger from the opposite hand to the Digital Crown of the device, completing a single-lead ECG. Patients held this position for 30 seconds and an ECG was recorded. This maneuver was either performed independently in children with neurodevelopmental abilities to do so, or with parental involvement as needed.

**Statistical analysis**. Quantitative data within the text are given as mean ± standard deviation. All numerical analysis was done using STATA version 15 (College Station, TX).

**Reporting summary**. Further information on research design is available in the Nature Portfolio Reporting Summary linked to this article.

## Results

**Patient characteristics**. The patient query is delineated in Fig. 1. A total of 145 documentations of *Apple Watch* were noted in patient records, with 41 unique patients identified that met study inclusion criteria with a confirmed arrhythmia identified (Table 1). The mean age at diagnosis was 13.8 ± 3.2 years with a

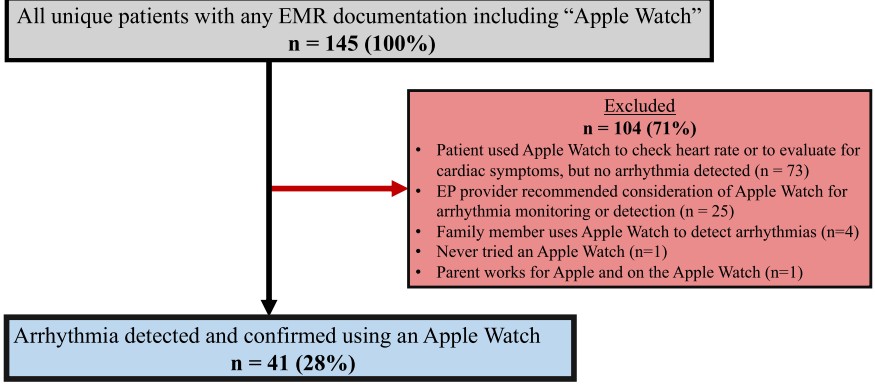

**Fig. 1 Patient selection and identification.** Flow diagram demonstrating the results of a query of the electronic-medical record system for the terms *Apple Watch*. EMR electronic-medical record.

**Table 1 Patient population and basic Apple Watch findings.**

| | |
|---|---|
| Patient population | N = 41 |
| Age at diagnosis (years) | 13.8 ± 3.2 |
| Female (n, %) | 27 (66%) |
| Weight at diagnosis (kg) | 57 ± 18 |
| Race (n, %) | |
| White, Caucasian | 25 (61%) |
| Hispanic, Latinx | 12 (29%) |
| Asian | 2 (5%) |
| Black, African American | 1 (2.5%) |
| Two or more races | 1 (2.5%) |
| Echocardiogram function (n, %) | |
| Normal | 40 (98%) |
| Moderately reduced | 1 (3%) |
| LVEF on echocardiogram (%) | 65 ± 5% |
| Congenital heart disease (n, %) | 5 (12%) |
| Invasive EP study performed (n, %) | 36 (88)% |
| Arrhythmia identified (n%) | |
| SVT | 36 (88%) |
| AVRT | 16 (44%) |
| AVNRT | 11 (30%) |
| EAT | 4 (11%) |
| Atrial Fibrillation | 2 (6%) |
| Atrial Flutter | 1 (3%) |
| No EPS (patient or family preference) | 2 (6%) |
| VT | 3 (7%) |
| Wide complex tachycardia | 1 (2.5%) |
| AV block with wide complex escape rhythm | 1 (2.5%) |
| Arrhythmia rate on Apple Watch during arrhythmia (bpm) | 220 [191–230] |
| Apple Watch role in arrhythmia diagnosis or management (n%) | |
| Establish initial diagnosis | 29 (71%) |
| New arrhythmia finding in a patient with a prior known diagnosis | 8 (19%) |
| Identified an episode of a previously known arrhythmia | 4 (10%) |
| Method of Apple Watch arrhythmia diagnosis (n, %) | |
| High heart rate notification with symptoms | 23 (56%) |
| Patient-triggered ECG recording during symptoms | 18 (44%) |

Patient sample characteristics, anatomical and electrophysiological cardiac diagnoses, and Apple Watch findings. Data are reported as either *mean ± standard deviation or median [interquartile range]*.
*kg* kilograms, *LVEF* left ventricular ejection fraction, *EP* electrophysiology, *SVT* supraventricular tachycardia, *AVRT* atrioventricular reentrant tachycardia, *AVNRT* atrioventricular nodal reentrant tachycardia, *EAT* ectopic atrial tachycardia, *VT* ventricular tachycardia, *AV* atrioventricular, *ECG* electrocardiogram.

**Table 2 Arrhythmias identified in patients with congenital heart disease (n = 5).**

| Pt # | Anatomic diagnosis | EP diagnosis | Apple Watch role | Treatment |
|---|---|---|---|---|
| 1 | TOF (neonatally repaired) | Monomorphic VT—RVOT VT | VT captured on ECG | Surgical ablation at time of PVR |
| 2 | ASD and VSD (closure in infancy) | SVT—Atypical AVNRT | Tachycardia alert with symptoms | Successful cryoablation |
| 3 | Outlet VSD (repair in infancy) | SVT—AVRT due to concealed left sided AP | SVT captured on ECG | Successful RF ablation |
| 4 | Valvar PS (mild) | SVT—Typical AVNRT | Tachycardia alert with symptoms | Successful cryoablation |
| 5 | Ebstein's Anomaly, LVNC (s/p OHT) | Atrial flutter | Atrial flutter captured on ECG | Successful cardioversion |

Anatomic and electrophysiologic diagnoses and outcomes specifically for patients with congenital heart disease in the patient sample.
*Pt* patient, *ID* identification number, *EP* electrophysiology, *TOF* Tetralogy of Fallot, *ASD* atrial septal defect, *VSD* ventricular septal defect, *PS* pulmonary stenosis, *LVNC* left ventricular noncompaction, *SVT* supraventricular tachycardia, *AVRT* atrioventricular reentrant tachycardia, *AVNRT* atrioventricular nodal reentrant tachycardia, *VT* ventricular tachycardia, *PVR* pulmonary valve repair, *RF* radiofrequency, *RVOT* right ventricular outflow tract.

mean weight at diagnosis of 57.2 ± 18.0 kg. The majority had normal ventricular function on echocardiography, though one patient (2%) had moderately depressed ventricular function at the time of presentation. Most patients had a structurally normal heart, though there were five patients (12%) with concomitant congenital heart disease (Table 2).

**Role of the Apple Watch**. Patients used their Apple Watch for arrhythmia characterization during symptomatic episodes either via a patient-initiated ECG using the Apple Watch ECG App (18 patients, 44%), or the high heart rate notification feature of the Apple Watch ECG App (23 patients, 56%). For 29 (71%) patients, the Apple Watch findings led the care team to pursue a workup resulting in a new arrhythmia diagnosis. In the remainder of cases, the Apple Watch captured an already known arrhythmia, or recurrence or progression of underlying arrhythmia substrate. In one patient with known congenital complete heart block and a narrow complex escape rhythm, the patient recorded an ECG

**Table 3 Cases of negative ambulatory monitor studies (n = 10).**

| Pt # | Duration of symptoms | Clinical scenario or symptoms | Monitors used (duration of use) | Apple Watch findings | EP study findings |
|---|---|---|---|---|---|
| 1 | 3 years | Palpitations | Holter monitor (48 h) | Tachycardia alert | AVRT |
| 2 | 2 years | Palpitations and chest pain | Patch monitor (7 days) | Tachycardia alert | AVNRT |
| 3 | 2 years | Palpitations with presyncope | Patch monitor (11 days) | Tachycardia alert | AVRT |
| 4 | 2 years | Exercise-induced palpitations and chest pain | Patch monitor (1 day × 2 trials, failed due to skin reaction to adhesive) | Triggered ECG showing SVT | AVNRT |
| 5 | 2 months | Palpitations and presyncope, intermittent WPW | Event monitor (30 days) | Triggered ECG showing SVT | WPW/AVRT |
| 6 | 7 years | Palpitations | Multiple event monitors of unknown wear time | Tachycardia alert | AVNRT |
| 7 | 1 year | Palpitations thought to be panic attacks, intermittent WPW | Multiple trials of patch monitors (3-, 4-, 6-, 8-, 10-, and 10-day monitoring periods) | Tachycardia alert | WPW/AVRT |
| 8 | 1 year | Palpitations | Patch monitor (14 days) | Tachycardia alert | AVNRT |
| 9 | 2 months | Palpitations with exertion | Patch monitor (1 day, terminated due to anxiety with device) | Triggered ECG showing SVT | No EPS—medical therapy |
| 10 | 6 years | Palpitations and chest tightness, thought to be due to asthma | Patch monitor (13 days) | Triggered ECG showing wide complex tachycardia | No EPS—medical therapy |

Cases in which patients underwent a workup involving a traditional ambulatory cardiac monitor which did not detect an episode of the patient's arrhythmia. The types of monitors used, the duration of use, and the role of the Apple Watch in connecting the patient to care is noted.
M male, F female, SVT supraventricular tachycardia, AVRT atrioventricular reentrant tachycardia, AVNRT atrioventricular nodal reentrant tachycardia, WPW Wolf-Parkinson-White Syndrome, ECG electrocardiogram, EP electrophysiology.

during symptoms of dizziness that demonstrated a brief slow wide complex escape rhythm.

Of patients who also underwent a workup involving a traditional ambulatory cardiac rhythm monitor, the traditional monitor did not detect an arrhythmia in 10 (29%) patients. These included cases in which patients wore a patch rhythm monitor, a Holter monitor, a 30-day event monitor, or a combination of any of the three (Table 3). Wear times varied by device, and many patients wore the monitor for greater than a week (up to a month). Two patients (6%) had documented skin reactions to the monitor adhesives.

Figure 2 shows several exemplary cases of triggered ECGs from the Apple Watch capturing arrhythmia events in children.

A total of 73 patients had documentation in the medical record of Apple Watch use for heart rate or cardiac symptom monitoring (Fig. 1) without ever having arrhythmias detected. In 18 of these patients (25%), abnormal heart rate or rhythm notifications from the Apple Watch principally led to them to seek pediatric cardiology care.

**Diagnostic results**. Formal arrhythmia diagnoses were made using standard, medically regulated diagnostic tools including clinic- or hospital-performed 12-lead ECGs, ambulatory patch rhythm monitors, or invasive EP study. Most patients had a diagnosis of supraventricular tachycardia (SVT) (36, 88%), with a small number with ventricular tachycardia (VT) (3, 7%), complete heart block (1, 2.5%), and one with a wide complex tachycardia (1, 2.5%) (Table 1). Of our patients with SVT, 34 (94%) underwent invasive EP study and were noted have atrioventricular reentrant tachycardia (AVRT) (16, 44%), atrioventricular nodal reentrant tachycardia (AVNRT) (11, 30%), ectopic atrial tachycardia (EAT) (4, 11%), atrial fibrillation (2, 6%), and atrial flutter (1, 3%). At the time of data collection, 2 (6%) had not undergone an EP study due patient or parental preference.

Overall, in the 36 (88%) patients that underwent invasive EP study, ablation or arrhythmia termination was successful in all but 4 (11%) patients. Of these four, two patients had documented SVT and 2 had documented VT. In the two patients with SVT, one had an ECG performed in the emergency department confirming the Apple Watch tracings of atrial fibrillation. This patient had an EP study for lone atrial fibrillation to assess for a concealed pathway, AVNRT, or an atrial tachycardia as the driver of their atrial fibrillation, though no substrate was noted as a cause of the atrial fibrillation. The other SVT patient was diagnosed with EAT based on Holter monitor results following SVT identification on Apple Watch ECG tracings but had no inducible EAT under anesthesia at EP study. In the two patients with VT, one was found to have RV inflow VT and had ablation deferred due to risk of conduction system injury due to the location of the VT focus in close proximity to the conduction system. The other VT patient with RV outflow tract VT underwent isthmus mapping and had a surgical ablation at the time of pulmonary valve replacement.

**Patients with congenital heart disease**. A listing of the patients with an Apple Watch identified arrhythmia and concomitant congenital heart disease is shown in Table 2. These patients included: Patient #1, a patient with a history of neonatal valve-sparing repair of Tetralogy of Fallot with a history of frequent palpitations. The palpitations led to a patient-triggered use of an Apple Watch ECG and the ECG tracing demonstrating VT was obtained. Patient #2, 3, and 4 developed symptoms of palpitations and had SVT captured on Apple Watch tracings or tachycardia alerts during symptomatic events and had SVT confirmed at EPS. Finally, Patient #5 underwent an orthotopic heart transplant in

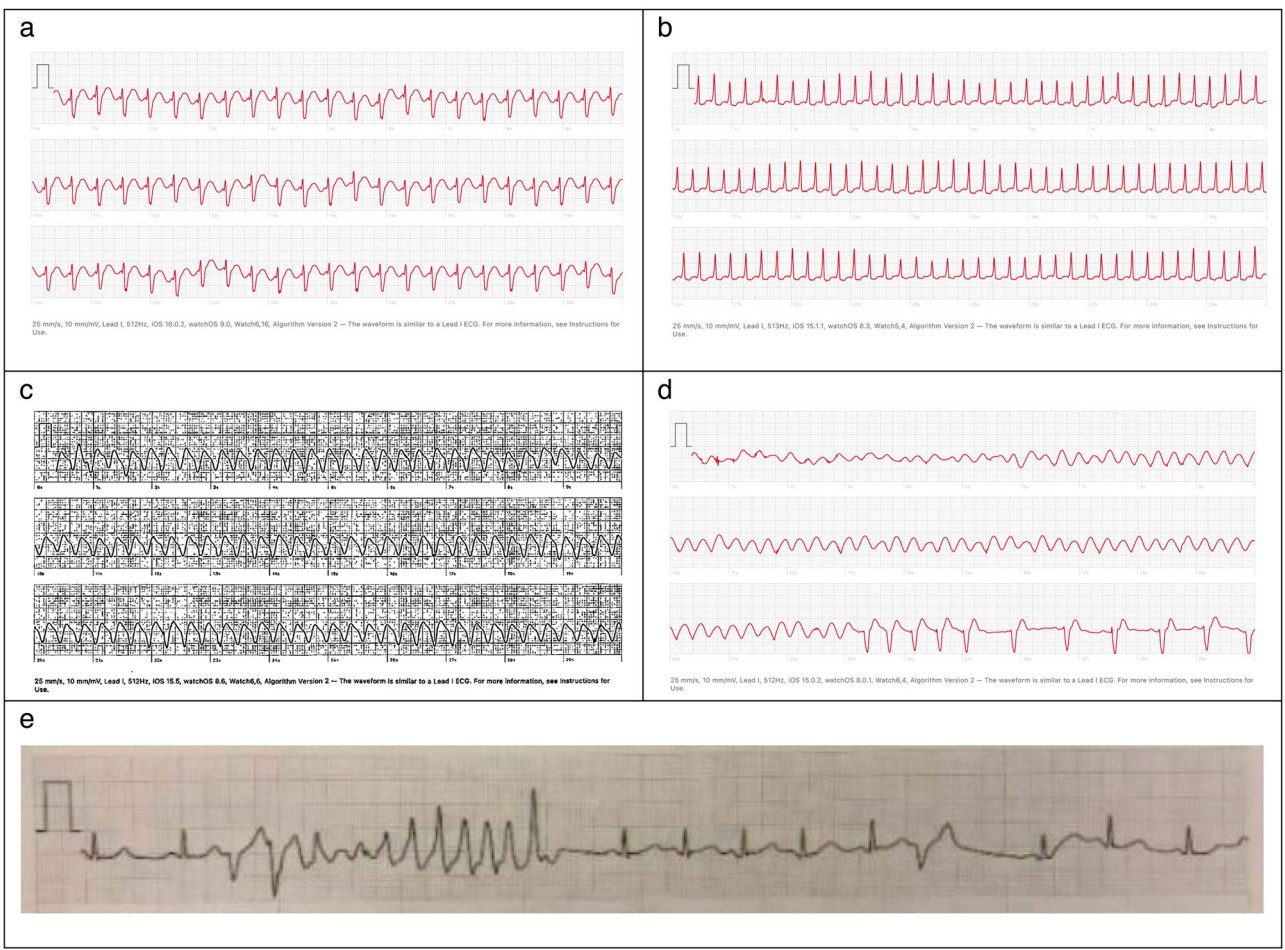

**Fig. 2 Featured cases of arrhythmias captured on Apple Watch.** Exemplary ECG tracings taken by patients using the Apple Watch. **a** A patient s/p OHT in infancy for Ebstein's anomaly with LVNC. The patient presented with palpitations and Apple Watch captured an episode of atrial flutter. **b** An otherwise healthy patient presenting with a 2-month history of racing heartbeat. The patient was unable to tolerate a Patch monitor for more than 1 day. Apple Watch captured an episode of SVT. **c** A patient with 6 years of palpitations and chest tightness, originally thought to be due to reactive airway disease. A patch monitor worn for 2 weeks was negative for any arrhythmias, even during symptomatic episodes. An exercise study was also normal. Apple watch recorded wide complex tachycardia during one of his episodes about 1 year later. **d** A patient with a history of neonatal valve-sparing repair of TOF with recurrent palpitations with exercise. During a symptomatic event an episode of monomorphic VT was recorded on the patient's apple watch, and further episodes were confirmed on a patch rhythm monitor. Further workup following this recording revealed RVOT VT. **e** A patient who was being evaluated for palpitations and ventricular bigeminy and trigeminy identified on ECG. The patient had patch rhythm monitor confirming bigeminy and trigeminy, but no tachycardia. Apple watch captured this episode of polymorphic ventricular tachycardia during a severe episode of palpitations. OHT orthotopic heart transplant, LVNC left ventricular noncompaction, SVT supraventricular tachycardia, TOF Tetralogy of Fallot, VT ventricular tachycardia, RVOT right ventricular outflow tract, ECG electrocardiogram.

infancy for Ebstein's anomaly with LVNC who had palpitations and an Apple Watch tracing confirmed recurrence of atrial flutter.

## Discussion
The rise in popularity of wearable health and fitness devices has prompted numerous evaluations of their roles in clinical practice. Data in adults have demonstrated that the Apple Watch can be an important diagnostic tool to help identify arrhythmias and decrease the time to symptomatic rhythm detection[7,8]. These studies have demonstrated that the Apple Watch can be considered for at-home cardiac rhythm monitoring in adults. Our investigation found that the Apple Watch can record arrhythmia events in children and may capture arrhythmia events that traditional ambulatory monitors do not. Thus, the Apple Watch can also play a relevant role in establishing a new arrhythmia diagnosis or arrhythmia surveillance in the pediatric population.

In contrast to the large studies of the Apple Watch in adult populations, data in children remain limited. Studies to date in various age groups and cardiac anatomy have concentrated on validating ECG measurements and heart rate sensor accuracy. Excellent agreement between the continuous pulse monitor of the Apple Watch and traditional monitor has been seen[14], though the sensor accuracy has been shown to be limited with heart rates over 210 beats per minute[14]. The consistency of the Apple Watch heart rate monitor in correlating with SVT events has been seen to be highly sensitive to the duration of the arrhythmia, particularly for events under 60 s long[13]. Thus, the sensor alone may have limitations with diagnosis of a tachyarrhythmias in children, making the ECG feature an especially relevant tool for arrhythmia characterization.

In pre-term neonates, Paech et al. showed excellent correlation of key intervals and amplitudes between lead I of a standard 12-lead ECG and an Apple Watch ECG placed on the neonate's wrist[12]. Similar results were previously obtained in children up to

16 years of age with and without congenital heart disease[10]. Notably, accuracy of the Apple Watch automated rhythm classification in children algorithm in these studies was poor. It is important to note that the Apple Watch ECG classification algorithm classifies recordings in the following categories: sinus rhythm, atrial fibrillation, low or high heart rate, inconclusive, or poor recording[15]. Notably missing in the classification algorithm are the tachyarrhythmias often seen in children, though as we have shown here, many children will use the high heart rate notification. The high hate rate notification algorithm, however, was designed for adults. For these reasons, in our practice the electrophysiologists did not rely on the Apple Watch ECG rhythm classification algorithm when interpreting ECG findings. Moreover, the Apple Watch ECG tracing feature is limited to patients 22 years or older, and pediatric patients in this series either modified their age to access the feature or used the device of an adult family member.

In this investigation, we evaluated the device's role in arrhythmia event identification in children. The Apple Watch made meaningful diagnostic contributions to patient care in our cohort. In 71% of cases, the Apple Watch findings prompted the team to pursue a workup which led to a new arrhythmia diagnosis. In general, patients in this cohort used the Apple Watch to corroborate their symptoms at home, prior to seeking the care of a cardiologist for palpitations. In 44% of cases, patients obtained an ECG at the time of their symptoms and brought these tracings to clinic. The ECG findings expedited our workup by providing objective evidence of arrhythmia in patients who would have otherwise undergone other ambulatory monitoring studies (Fig. 2). Such cases are becoming more common in the published literature as well, as the use of wearable devices in children is becoming increasingly widespread. Reports of the Apple Watch detecting episodes of SVT in children and adolescents are most common, and others have used it in cases where traditional monitors failed to detect an arrhythmia[16,17].

The remainder of our patients (56%) correlated their symptoms with heart rate alerts on the Apple Watch. Although less specific of a finding than an ECG showing evidence of arrhythmia, instances of tachycardia well beyond the age-adjusted normal range limits were helpful to patients in validating the subjectivity of their symptoms. Optical heart rate sensors are additionally quite common in other wearable devices, meaning that a variety of wearable devices can be useful to this end as well. The concept of passive heart rate monitoring has been a rich topic of discussion in the literature surrounding pediatric cardiac event monitors[18].

In the cases where the Apple Watch didn't help establish a new diagnosis, the Apple Watch was still valuable in identifying relapse or progression of a previously known arrhythmia. In many cases, patients would specifically wear an Apple Watch for symptom tracking following their initial diagnosis. There were several cases of patients who had previously had acutely successful ablations for SVT, who were brought to care for further treatment based on recurrence of SVT identified on the Apple Watch via the heart rate monitor and/or patient-initiated ECGs.

A concern often raised by healthcare providers regarding the use of wearable devices such as the Apple Watch is the false-positive rate of device-generated alerts. Especially since most wearable consumer electronic devices such as the Apple Watch are not designed for use in children, many of the automated features may incorrectly alert pediatric patients of heart rate or rhythm abnormalities, in some cases prompting families to seek unnecessary medical care. In our study, of the patients without arrhythmia whose medical documentation indicated that the Apple Watch was used for recreational cardiac monitoring, we found that 25% presented to care principally due to high heart rate or irregular heart rhythm alerts on the Apple Watch. Although our study did not have simultaneous patch rhythm monitor data on these patients to confirm or refute a possible arrhythmia associated with these alerts, it is notable that a substantial portion of unnecessary healthcare visits were likely a result of these alerts.

The Apple Watch was also able to identify an arrhythmia when traditional ambulatory ECG monitors were not. Traditional monitors are hampered by limited wear time, patient non-compliance secondary to skin sensitivity to the monitor adhesives, and/or the sporadic and infrequent nature of symptoms. In a prospective, randomized comparison of one hundred patients undergoing extended cardiac monitoring for syncope or pre-syncope using Holter monitors and implantable loop recorders, only 39% of rhythm-symptom correlations were achieved in the first two weeks of monitoring[19]. Another study of the Zio Patch Rhythm Monitor showed an SVT detection rate of only 52% after two weeks of monitoring in patients suspected to have arrhythmia[20]. There is clearly a need for a longer-term non-invasive method to perform extended cardiac rhythm monitoring in children beyond the capabilities of the currently available medically-issued cardiac rhythm monitors. As is the case with most medical technologies, wearable ECG-capable devices are often limited to the adult population, limiting the diagnostic options for pediatric patients. Our study attempts to expand these options for children and shows that the Apple Watch is a wearable technology that can play a valuable role in pediatric arrhythmia characterization.

**Limitations**. This is a retrospective case series, and not a randomized trial, which carries some limitations. This retrospective chart review study was not able to characterize the true false-positive rates of Apple Watch findings in children. While we were able to conclude that some patients sought testing or cardiac consultation at our center based principally on abnormal heart rate or rhythm notifications, our study can only conclude that these findings were false-positive alerts by virtue of the patient not receiving an eventual arrhythmia diagnosis. Specifically, we could not concurrently compare the Apple Watch findings to gold standard ambulatory methods, which limits our ability to comment on the accuracy of the Apple Watch heart rate or rhythm notifications. Furthermore, because we do not have the same level of objective testing for the excluded patients, we cannot state with certainty that they did not have a true arrhythmia. For these reasons, this investigation focused on the verifiable diagnostic contributions from the Apple Watch in cases of confirmed pediatric arrhythmias and our ability to quantify the predictive value of the Apple Watch to establish a correct diagnosis was limited.

The cohort of patients in this analysis are also from a large referral center, and there is likely some degree of selection bias, with a higher degree of patients referred to our center with true arrhythmias detected. The retrospective nature of this study increases the yield of captured arrhythmias and is not representative of the number of alerts that patients of a broader population might receive which are not presented to a healthcare provider. Additionally, 29% of patients in our study had a previously known arrhythmia, further increasing our yield of captured arrhythmias. As data was obtained from the EMR, it is also possible that there were some patients that used an Apple Watch, but we could not identify them via a retrospective EMR assessment. Further studies are needed to assess how often inconsequential care is sought for abnormal heart rate or rhythm notifications on an Apple Watch compared to patients presenting due to symptoms alone. While we have shown that the Apple

Watch successfully identified instances of arrhythmia leading to clinical testing and diagnoses, it is difficult to fully characterize the diagnostic utility of the Apple Watch for all children with cardiac symptoms. We could also not determine how many total tracings were reviewed by the electrophysiologist team over the study period. Lastly, our study involved patients who had access to a personally owned Apple Watch, biasing our sample to those who would have the interest and the means to obtain and use such a device. Further studies should be performed to assess the device use in a wider population base.

## Conclusions

The Apple Watch can record arrhythmia events in some children. For many patients in this cohort, the Apple Watch findings initiated a workup resulting a new arrhythmia diagnosis, and in some, it recorded arrhythmias that traditional ambulatory monitors did not. This data demonstrates that consumer wearables such as the Apple Watch may play an important role in arrhythmia diagnosis and surveillance in children. Further studies prospectively comparing the Apple Watch with traditional ambulatory cardiac rhythm monitors are needed to best characterize its diagnostic value in children and further understand how to integrate patient-submitted ambulatory ECG data into clinical practice.

## Data availability

All data used to generate the results and figures are located in controlled access data storage at Stanford University. The datasets generated and analyzed during the current study are not publicly available to protect patient privacy but are available from the corresponding author on reasonable request.

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

## Acknowledgements

We would like to thank Stanford Biodesign Digital Health students Abhinav Agarwal, Sidharth Gopisetty, Riya Karumanchi, Caitlin Kunchur, Danny Park, Shriya Reddy, Julia Rhee, Raghav Samavedam, Varun Shenoy, and Ananya Vasireddy for their work as part of our Pediatric Apple Watch Study team. Apple Watch is a trademark of Apple Inc., registered in the USA and other countries and regions.

## Author contributions

A.Z. and S.R.C. conceived the study. A.Z., H.C., H.G., A.M.D., K.S.M., and W.G. designed the study. N.K.B., A.T., E.L., and B.N. collected the data. A.Z. and S.R.C. analyzed the data. A.Z. and S.R.C. prepared the manuscript. A.Z., H.C., H.G., A.M.D., A.T., K.S.M., W.G., V.R., P.S., V.B., O.A., X.B.L., M.P., and S.R.C. critically reviewed the manuscript.

## Competing interests

M.P. is a consultant for Apple Inc., and Boston Scientific Inc., has a <1% equity stake in QALY Inc., and receives research support from Apple Inc., the NIH and the NHLBI. S.C. has received research support from Apple Inc. for separate work.
