## [Peer Review File · Communications Medicine]

Reviewers' comments:

Reviewer #1 (Remarks to the Author):

This is a wonderful case series on the utility of the Apple Watch in the diagnosis of cardiac arrhythmias in a paediatric population. 41 patients were identified from a medical record "biopsy" of 145 electronic medical records after searching for the term "Apple watch". There were 37 arrhythmias confirmed including 36 SVT (88%), 3 VT (7%), 1 heart block (2.5%) and wide 1 complex tachycardia (2.5%). The remarkable findings were that an invasive EP study confirmed diagnosis in 34 of the 36 patients (94%) with SVT (2 non-inducible) and that the the Apple Watch findings prompted a workup resulting in a new arrhythmia diagnosis for 29 patients (71%). Remarkably, traditional ambulatory cardiac monitors were also worn by 35 patients (85%), with traditional monitors failing to detect arrhythmia in 10 (29%) patients. The authors conclude that the Apple Watch can play an important role in pediatric arrhythmia diagnosis.

I have the following comments:

1. It is unclear what series of Apple Watch was used, please specify.
2. The methodology of how Apple watch records an ECG is important to describe in the methods.
3. Please list the duration of Holter monitoring.
4. please acknowledge the limitation that this is a retrospective evaluation of patients, hence there may be a significant bias, and the yield might be much lower when applied to a broader population.
5. Please also acknowledge the limitation that ~30% already had a known arrhythmia diagnosis, hence this population is "primed" to get a higher yield.
6. How many transmissions were reviewed from the Apple Watch?
7. The ECG examples are very informative.
8. Please acknowledge that this is not a randomised trial.

Reviewer #2 (Remarks to the Author):

The authors have submitted a manuscript intending to study the utility of the Apple Watch on arrhythmia detection in children.

The manuscript reads well and provides a comprehensive review of the background of heart rate monitoring in children and the technical need for better tools.

The authors may want to consider the following in the manuscript :

Background:

Nicely written; please consider introducing the concept of a high heart rate alert and a single lead ECG as two different functionalities of the same device.

Methods :

It could have been clearer if the caregivers were asked to purchase an Apple watch or if it was voluntary.

Were consecutive patients included? How was the Apple watch documented (every patient was asked about the device ?)

Readers would also like to know if the children performed the ECGs or if they asked for family members to help them , as there may be limitations associated with children activating a device based on their physical and neurocognitive levels of functioning.

More information is needed if the device was worn every day and the duration of use (continuous versus PRN use)

Results :

The readers would benefit from understanding how ventricular tachycardia was diagnosed with a single lead Apple watch trace ;

Other explanations regarding how a diagnosis of atrial flutter was made on a narrow complex tachycardia (Fig 2) would educate readers.

Similarly, a tracing suggestive of polymorphic VT (15 yr boy with neonatal valve-sparing TOF repair) is reported as a monomorphic VT, were these tracings adjudicated by an Electrophysiologist?

Discussion :

Heart rate notification rather than ECG diagnosis was in 56% of the study population; therefore, any other wearable that can detect the heart rate rather than ECG may also be useful. Some discussion around this topic will be helpful.

Conclusion :

The conclusion may be modified as the data shows that the Apple Watch can record clinically significant arrhythmia in some (but not all children)

Responses to the Reviewers

Reviewer comments are reproduced below. Author responses are indented and italicized. Please note that line numbers referring to additions correspond to the copy with track-changes enabled.

Reviewer #1 Comments:

This is a wonderful case series on the utility of the Apple Watch in the diagnosis of cardiac arrhythmias in a paediatric population. 41 patients were identified from a medical record "biopsy" of 145 electronic medical records after searching for the term "Apple watch". There were 37 arrhythmias confirmed including 36 SVT (88%), 3 VT (7%), 1 heart block (2.5%) and wide 1 complex tachycardia (2.5%). The remarkable findings were that an invasive EP study confirmed diagnosis in 34 of the 36 patients (94%) with SVT (2 non-inducible) and that the Apple Watch findings prompted a workup resulting in a new arrhythmia diagnosis for 29 patients (71%). Remarkably, traditional ambulatory cardiac monitors were also worn by 35 patients (85%), with traditional monitors failing to detect arrhythmia in 10 (29%) patients. The authors conclude that the Apple Watch can play an important role in pediatric arrhythmia diagnosis.

I have the following comments:

1. It is unclear what series of Apple Watch was used, please specify.

Thank you for this important point. The exact model of each individual patient's Apple Watch device was not uniformly solicited as a part of our study. We added text to the methods section describing the sensors in the Apple Watch that contributed to the types of data collected in our study (lines 114-116). We additionally specified that the ECG sensor was not integrated into the Apple Watch until Series 4, and all patients in this series that had ECG tracings reviewed were all Series 4 or later.

2. The methodology of how Apple watch records an ECG is important to describe in the methods.

Adding to our response to the above comment, we added text describing how an ECG is collected using the Apple Watch (lines 116-119).

3. Please list the duration of Holter monitoring.

Thank you for highlighting this. We agree that it is particularly impactful to report the individual monitoring periods of Holter, patch rhythm, and other event monitors in the cases where ambulatory methods did not capture arrhythmia. This data is documented and included in table 3.

4. Please acknowledge the limitation that this is a retrospective evaluation of patients, hence there may be a significant bias, and the yield might be much lower when applied to a broader population.

Thank you for this excellent point. We have included text addressing this in the Limitations section (lines 298-300).

5. Please also acknowledge the limitation that ~30% already had a known arrhythmia diagnosis, hence this population is "primed" to get a higher yield.

We would agree with that important point, and it has been added to the Limitations section (lines 300-302).

6. How many transmissions were reviewed from the Apple Watch?

We also thought about trying to determine exactly how many transmissions were reviewed by each electrophysiologist. Unfortunately, due to the retrospective nature of the study, we do not have a way of retrospectively determining how many total tracings were reviewed. Our clinical documentation did not uniformly capture how many transmissions were received per

patient. We have added this point to the Limitations section of manuscript (lines 309-310). While we could not assess the total number of tracings, we were able to assess the role of the Apple Watch in bringing each patient to care to try to get a better sense of the impact of watch usage on care utilization (See Role of the Apple Watch in Results). Our hope is that this captures the false-positive rate of Apple Watch alerts on a clinical level, since it provides a measure of how many clinical encounters driven by the Apple Watch resulted in a diagnosed arrhythmia.

7. The ECG examples are very informative.

Thank you, we found them quite interesting as well we are delighted to know that they were helpful.

8. Please acknowledge that this is not a randomized trial.

Text was added to explicitly mention this in the Limitations section (lines 283-284).

Reviewer #2 Comments:

The authors have submitted a manuscript intending to study the utility of the Apple Watch on arrhythmia detection in children.

The manuscript reads well and provides a comprehensive review of the background of heart rate monitoring in children and the technical need for better tools.

The authors may want to consider the following in the manuscript:

Background:

1. Nicely written; please consider introducing the concept of a high heart rate alert and a single lead ECG as two different functionalities of the same device.

Thank you for this important piece of context. It has been added to the background section (lines 92-94).

Methods:

2. It could have been clearer if the caregivers were asked to purchase an Apple watch or if it was voluntary.

Thank you for this important consideration. Submission of data was voluntary, and all data originated from patient- or caregiver-owned Apple Watch devices. This is clarified in the methods section (lines 105-107).

3. Were consecutive patients included? How was the Apple watch documented (every patient was asked about the device?)

The role of the Apple Watch was documented based on the information that the patient self-presented to the provider. Providers did not specifically solicit data from patients' Apple Watches. As this was a retrospective analysis, we queried our electronic medical record database for mentions of the Apple Watch to then assess the device's involvement in patient care. The reviewer brings up an important additional limitation, that it is possible that patients used an Apple Watch, and that information was not included in the electronic medical record. This was added to the Limitations section (lines 302-304).

4. Readers would also like to know if the children performed the ECGs or if they asked for family members to help them, as there may be limitations associated with children activating a device based on their physical and neurocognitive levels of functioning.

This is a particularly important point as we think about the role of wearable devices in children of various neurodevelopmental and physical levels of functioning. While we do not have the exact distribution of which patients performed the ECG recording maneuver independently versus with help, we acknowledged this important nuance in the methods section (lines 120-121). While we don't have the data on this due to the retrospective nature of the investigation, in our experience, the older patients (i.e. teenagers) largely obtain tracings on their own, while the younger patients often had the help of a family member.

5. More information is needed if the device was worn every day and the duration of use (continuous versus PRN use).

We do not have definitive data on whether the device was worn continuously or as needed, however, the two cardiac monitoring features of the Apple Watch carry unique usage implications which we highlighted in the methods section. The optical heart rate sensor allows for continuous measurements and passive alerts, while the ECG involves patient-initiated recordings. We have highlighted this point in lines 114-119. Thank you for highlighting this important usage point.

Results :

6. The readers would benefit from understanding how ventricular tachycardia was diagnosed with a single lead Apple watch trace ; Other explanations regarding how a diagnosis of atrial flutter was made on a narrow complex tachycardia (Fig 2) would educate readers. Similarly, a tracing suggestive of polymorphic VT (15 yr boy with neonatal valve-sparing TOF repair) is reported as a monomorphic VT, were these tracings adjudicated by an Electrophysiologist?

Thank you for this important point—no arrhythmia diagnosis was made using the single-lead Apple Watch tracing on its own. In each situation the Apple Watch tracings were reviewed by an electrophysiologist and the electrophysiologist was suspicious of the diagnoses, but the Apple Watch recordings were confirmed using standard, medically regulated diagnostic methods to ensure an accurate diagnosis. This was clarified in the manuscript in lines 161-163 in the results section and lines 107-109 in the methods section.

Discussion:

7. Heart rate notification rather than ECG diagnosis was in 56% of the study population; therefore, any other wearable that can detect the heart rate rather than ECG may also be useful. Some discussion around this topic will be helpful.

We would absolutely agree with the reviewer, we added text highlighting this in the discussion section, along with an added reference (lines 244-247).

Conclusion:

8. The conclusion may be modified as the data shows that the Apple Watch can record clinically significant arrhythmia in some (but not all children)

Thank you for this distinction. It has been added in the conclusion section (line 315).

REVIEWERS' COMMENTS:

Reviewer #1 (Remarks to the Author):

The authors have addressed my questions satisfactorily.

Reviewer #2 (Remarks to the Author):

The authors have made significant positive changes to the manuscript and addressed the comments that were made. As a result, the manuscript reads very well. Congratulations!